# Uncovering Surprising Behaviors in Reinforcement Learning via Worst-case Analysis

## Abstract

Reinforcement learning agents are typically trained and evaluated according to their performance averaged over some distribution of environment settings. But does the distribution over environment settings contain important biases, and do these lead to agents that fail in certain cases despite high average-case performance? In this work, we consider worst-case analysis of agents over environment settings in order to detect whether there are directions in which agents may have failed to generalize. Specifically, we consider a 3D first-person task where agents must navigate procedurally generated mazes, and where reinforcement learning agents have recently achieved human-level average-case performance. By optimizing over the structure of mazes, we find that agents can suffer from catastrophic failures, failing to find the goal even on surprisingly simple mazes, despite their impressive average-case performance. Additionally, we find that these failures transfer between different agents and even significantly different architectures. We believe our findings highlight an important role for worst-case analysis in identifying whether there are directions in which agents have failed to generalize. Our hope is that the ability to automatically identify failures of generalization will facilitate development of more general and robust agents. To this end, we report initial results on enriching training with settings causing failure.

## 1 Introduction

Reinforcement Learning (RL) methods have achieved great success over the past few years, achieving human-level performance on a range of tasks such as Atari (Mnih et al., 2015), Go (Silver et al., 2016), Labyrinth (Espeholt et al., 2018), and Capture the Flag (Jaderberg et al., 2018).

On these tasks, and more generally in reinforcement learning, agents are typically trained and evaluated using their average reward over environment settings as the measure of performance, i.e.

$$\mathbb{E}_{P(e)}\left[R(\pi(\theta), e)\right],$$

where $\pi(\theta)$ denotes a policy with parameters $\theta$, $R$ denotes the total reward the policy receives over the course of an episode, and $e$ denotes environment settings such as maze structure in a navigation task, appearance of objects in the environment, or even the physical rules governing environment dynamics. But what biases does the distribution $P(e)$ contain, and what biases, or failures to generalize, do these induce in the strategies agents learn?

To help uncover biases in the training distribution and in the strategies that agents learn, we propose evaluating the *worst-case* performance of agents over environment settings, i.e.

$$\min_{e \in \mathcal{E}} \mathbb{E}\left[R(\pi(\theta), e)\right],$$

where $\mathcal{E}$ is some set of possible environment settings.

Worst-case analysis can provide an important tool for understanding robustness and generalization in RL agents. For example, it can help us with:

- **Understanding biases in training** Catastrophic failures can help reveal situations that are rare enough during training that the agent does not learn a strategy that is general enough to cope with them.

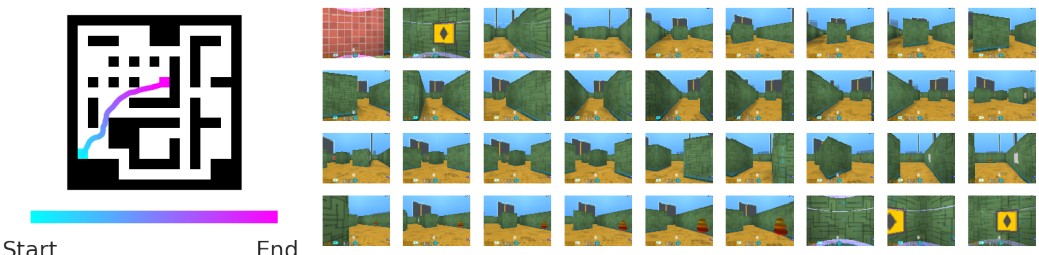

Start        End

Figure 1: **Navigation task.** (left) Example maze from the training distribution together with the path taken by the agent from spawn (cyan) to goal (magenta). (right) Frames from top left to bottom right correspond to agent observations as it takes the path from spawn to goal. Note that while the navigation task may look simple given a top down view, the agent only receives very partial information about the maze at every step, making navigation a difficult task.

- **Robustness** For critical systems, one would want to eliminate, or at least greatly reduce, the probability of extreme failures.
- **Limiting exploitability** If agents have learned strategies that fail to generalize to particular environment settings, then an adversary could try and exploit an agent by trying to engineer such environment settings leading to agent failure.

In this work, we use worst-case analysis to investigate the performance of a state-of-the-art agent in solving a first-person 3D navigation task; a task on which agents have recently achieved average-case human-level performance (Wayne et al., 2018). By optimizing mazes to minimize the performance of agents, we discover the existence of mazes where agents repeatedly fail to find the goal (which we refer to as a *Catastrophic Failure*).

**Our Contributions**    To summarize, the key contributions of this paper are as follows:

1. We introduce an effective and intuitive approach for finding simple environment settings leading to failure (Section 2).
2. We show that state-of-the-art agents carrying out navigation tasks suffer from drastic and often surprising failure cases (Sections 3.1 and 3.2).
3. We demonstrate that mazes leading to failure transfer across agents with different hyperparameters and, notably, even different architectures (Section 3.3).
4. We present an initial investigation into how the training distribution can be adapted by incorporating adversarial and out-of-distribution examples (Section 4).

## 2  APPROACH

**Tasks**    We consider agents carrying out first-person 3D navigation tasks. Navigation is of central importance in RL research as it captures the challenges posed by partially observable Markov decision processes (POMDPs). The navigation tasks we use are implemented in DeepMind Lab (DM Lab) (Beattie et al., 2016). [1] Each episode is played on a $15 \times 15$ maze where each position in the maze may contain a wall, an agent spawn point, or a goal spawn point. The maze itself is procedurally generated every episode, along with the goal and agent spawn locations. The goal location remains fixed throughout an episode, while the agent spawn location can vary. In training, the agent respawns at different locations each time they reach the goal, while for our optimization and analysis we limit the agent to the same spawn location. Agents receive RGB observations of size $96 \times 72$ pixels, examples of which are provided in Figure 1. Episodes last for 120 seconds and

---

[1]A full description and code for the tasks can be found at `https://github.com/deepmind/lab/tree/master/game_scripts/levels/contributed/dmlab30#goal-locations-large`.

are played at a framerate of 15 frames per second. The agent receives a positive reward of 10 every time it reaches the goal, and 0 otherwise. On this specific navigation task, RL agents have recently achieved human-level average-case performance (Wayne et al., 2018).

**Agents**   We perform our analysis on Importance Weighted Actor-Learner Architecture agents trained to achieve human-level average-case performance on navigation tasks. These agents can be described as async batched-a2c agents with the V-trace algorithm for off policy-correction, and we henceforth refer to these as A2CV agents (Espeholt et al., 2018). Details of the training procedure are provided in Appendix A.1.

**Search Algorithm**   If we are interested in worst-case performance of agents, how can we find environment settings leading to the worst performance? In supervised learning, one typically uses gradient based methods to find inputs that lead to undesired output (Biggio et al., 2013; Szegedy et al., 2013; Goodfellow et al., 2014). In contrast, we search for environment settings leading to an undesired outcome at the end of an episode. This presents a challenge as the environment rendering and MDP are not differentiable. We are therefore limited to black-box methods where we can only query agent performance given environment settings.

To search for environment settings which cause catastrophic failures, we propose the local search procedure described in Algorithm 1 (visualizing the process in Figure 2). Concretely, we generate a set of initial candidate mazes by sampling mazes from the training distribution. We then use the `Modify` function on the maze which yielded the lowest agent score to randomly move two walls to produce a new set of candidates, rejecting wall moves that lead to unsolvable mazes. Importantly, this method is able to effectively find catastrophic failure cases (as we demonstrate in Section 3.1), while also having the advantage of being intuitive to understand and implement.

---

**input** : num_iterations, num_candidates, num_evaluations and function `Modify`
**output:** An environment setting best

candidates ← `GenerateCandidates`(num_candidates);
**for** $i \leftarrow 1$ **to** num_iterations **do**
    best ← `Evaluate`(candidates, num_evaluations);
    candidates ← `Modify`(best, num_candidates);
**end**
best ← `Evaluate`(candidates, num_evaluations);

**Algorithm 1:** Method for finding environment settings leading to failure cases.

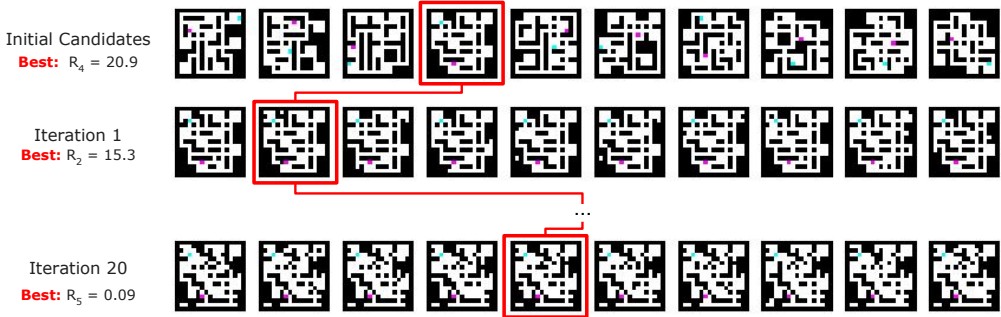

Figure 2: **Example of search procedure**. First, we generate a set of 10 initial candidate mazes by sampling from the training distribution. We then `Evaluate` each with the agent over 30 episodes, select the best maze (i.e. lowest agent score), and `Modify` this maze by randomly moving two walls to form the next set of candidates (Iteration 1). This process is repeated for 20 iterations, leading to a maze where the agent score is 0.09 in this example (i.e. the agent finds the goal once in 11 episodes). In Appendix A.2.1 we detail the computational requirements of this search procedure.

## 3   EXPERIMENTS

The agents we study achieve impressive average-case performance, but how much does their worst-case performance differ from their average-case performance? To investigate this, we consider the worst-case performance over a large set of mazes, including mazes that are not possible under the training distribution.

A natural question to ask is whether *any* departure from the wall structure present during training will lead to agent failure. To test this, we evaluate the agent on samples from a distribution of mazes containing all mazes agents could be evaluated on during the search. In particular, we randomly select agent and goal spawn locations in the first step and then randomly move 40 walls, corresponding to the same actions taken by our optimization procedure, but where the actions are chosen randomly rather than in order to minimize agent performance. We find that agents *do* generalize to random mazes from the set we consider. In fact, we find that the average score obtained by agents on randomly perturbed mazes is slightly *higher* than on the training distribution, with agents obtaining an average of 45 goal reaches per two minute episode. The increased performance is likely due to the agent spawn location being fixed, making it easier for the agent to return to the goal once found.

The considered agents generalize in the sense that agent performance is not reduced on average by out-of-distribution wall structure. But what about the worst case over all wall structures? Have the agents learned a general navigation strategy that works for all solvable mazes? Or do there exist environment settings that lead to catastrophic failures with high probability? In this section, we investigate these questions. We define a *Catastrophic Failure* to be an agent failing to find the goal in a two minute episode (1800 steps). As detailed below, we find that not only do there exist mazes leading to catastrophic failure, there exist surprisingly simple mazes that lead to catastrophic failure for agents yet are consistently and often rapidly solved by humans.

### 3.1   RESULT 1: CATASTROPHIC FAILURES EXIST

Do environment settings leading to catastrophic failure exist for the agents we are considering? By searching over mazes using the procedure outlined in Algorithm 1, we find mazes where agents fail to find the goal on many episodes, only finding the goal 10% of the time. In fact, some individual mazes lead to failure across five different agents we tested, with even the best performing agent only finding the goal in 20% of the episodes.

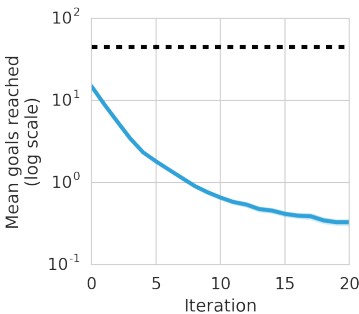
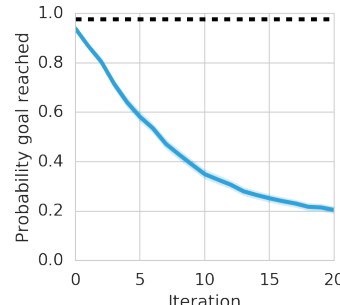

(a) Average number of goals reached per episode over the course of the optimization.

(b) Probability of the agent reaching the goal in an episode.

Figure 3: **The search algorithm is able to rapidly find mazes where agents fail to find the goal.** (a) The objective used for the optimizer is average agent score. The dashed line corresponds to average goals reached on randomly perturbed mazes. (b) Minimizing score also leads to a low probability of at least one goal retrieval in an episode. The dashed line corresponds to average probability of reaching a goal on randomly perturbed mazes. The blue lines are computed by averaging across 50 optimization runs.

Optimization curves for our search procedure are given in Figure 3. Note that while we define catastrophic failure as failure to find the goal, the actual objective used for the optimization was average number of goal reaches during an episode. Using average number of goals gives a stronger

signal at the start of the optimization process. Finding mazes leading to lower average number of captures is easier than finding mazes where the agent rarely finds the goal even once. As can be seen, despite finding the goal on average 45 times per episode on randomly perturbed mazes, on mazes optimized to reduce score, agents find the goal on average only 0.33 times per episode, more than a $100\times$ decrease in performance. In terms of probability of catastrophic failure, we note that despite agents finding the goal in approximately 98% of episodes on randomly perturbed mazes, using our method, on average we find mazes where agents only finds the goal in 30% of the episodes.

Example trajectories agents take during failures are visualized in Figure 4. The trajectories often seem to demonstrate a failure to use memory to efficiently explore the maze with the agent repeatedly visiting the same locations multiple times.

The mazes presented in Figure 4 appear to be of higher complexity than mazes seen during training. This suggests that to obtain agents that truly master navigation, more complex mazes should be included in the training distribution. However, we can ask whether it is only more complex mazes that lead to catastrophic failure or whether there are also simple mazes leading to catastrophic failure. This is a question we explore in the next subsection.

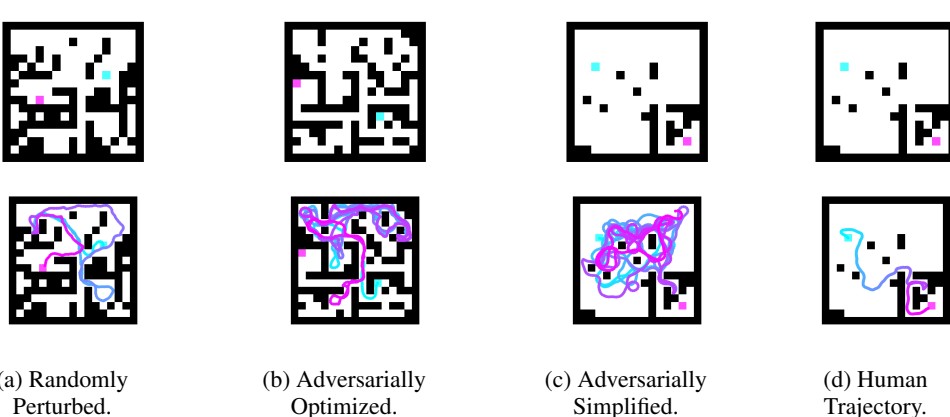

| (a) Randomly Perturbed. | (b) Adversarially Optimized. | (c) Adversarially Simplified. | (d) Human Trajectory. |

Figure 4: **Example mazes leading to low scores and example trajectories on these mazes.** (a) Maze with randomly perturbed walls. Despite being out of distribution, agents find the goal in 98% of episodes on such mazes and are able to get the goal on average 45 times per episode. (b) Maze obtained after 20 iterations, moving two walls at each iteration to minimize reward. All agents find the goal on such mazes in less than 20% of episodes. (c) Maze obtained through additional iterations of removing walls. All agents find the goal on such mazes in less than 40% of episodes. (d) Human trajectory on the same maze as in (c). Humans are able to consistently find the goal on such mazes.

## 3.2    RESULT 2: SIMPLE CATASTROPHIC FAILURES EXIST

While the existence of catastrophic failures may be intriguing and perhaps troubling, one might suspect that the failures are caused by the increased complexity of the mazes leading to failure relative to the mazes the agent is exposed to during training, e.g., the mazes leading to failure contain more dead ends and sometimes have lower visibility. Further, understanding the cause of failure in such mazes seems challenging due to the large number of wall structures that may be causing the agent to fail. In this section, we explore whether there exist *simple mazes* which lead to catastrophic failures. As our measure of complexity, we use the total number of walls in the maze. We also evaluate humans on such mazes to get a quantitative measure of maze complexity.

To find simple mazes which lead to failure, we first follow the same procedure as in the previous section, producing a set of mazes which all lead to catastrophic failures (i.e. a low agent scores). Next, we use this set of mazes as the initial set of candidates in our search algorithm, however we now use a `Modify` function that removes a single randomly chosen wall each iteration. This process is repeated for 70 iterations, searching for a maze with few walls while maintaining low agent score.

In Figure 4, we present the resulting simple mazes and the corresponding agent trajectories from our optimization procedure. Interestingly, we find that one can remove a majority of the walls in a maze

and still maintain the catastrophic failure (i.e. very low agent score). Of note is that a number of these mazes are strikingly simple, suggesting that there exists structure in the environment that the agent has not generalized to. For example, we can see that placing the goal in a small room in an otherwise open maze can significantly reduce the agent's ability to find the goal.

**Human baselines** While these simple maps may lead to catastrophic failure, it is unclear whether this is because of the agent or whether the maze is difficult in a way that is not obvious. To investigate this, we perform human experiments by tasking humans to play on a set of 10 simplified mazes.

Notably, we find that human players are able to always locate the goal in every maze and typically within one third of the full episode length. This demonstrates that the mazes are comfortably solvable within the course of an episode by players with a general navigation strategy. We provide a detailed comparison of agent and human performance in Appendix A.3.

**Analysis** One question that may arise is the extent to which these mazes are isolated points in the space of mazes. That is, if the maze was changed slightly, would it no longer lead to catastrophic failure? To test this, we investigate how sensitive our discovered failure mazes are with respect to the agent and goal spawn locations on simplified adversarial mazes. As can be seen in Figure 5, we find that for a large range of spawn locations, the mazes still lead to failure. This indicates that there is specific local maze structure which causes agents to fail.

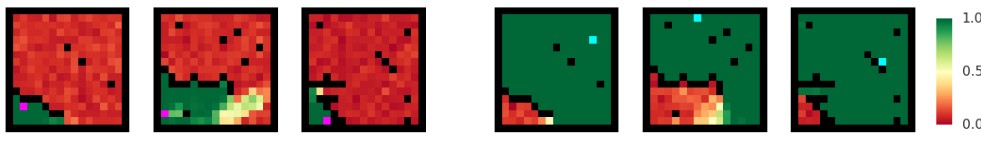

(a) Agent locations, with goal (magenta) fixed.          (b) Goal locations, with agent (cyan) fixed.

Figure 5: **Adversarial mazes are robust to change of spawn positions.** The probability of goal retrieval (shown with the color bar) remains low across large portions of the simplified maze as the (a) agent spawn locations and (b) goal locations are moved for each episode..

Procedures for finding such simple mazes may prove useful as a tool for debugging agents and understanding the ways in which training has led them to develop narrow strategies that are good enough for achieving high average-case performance.

### 3.3 RESULT 3: FAILURE MAZES TRANSFER ACROSS AGENTS

We have found failure cases for individual agents, but to what extent do these failure cases highlight a specific peculiarity of the individual agent versus a more general failure of a certain class of agents, or even a shortcoming of the distribution used for training? In this section, we investigate whether mazes which cause one trained agent to fail also cause other agents to fail.

We consider two types of transfer: (1) between different hyperparameters of the same model architecture, and (2) between different model architectures. To test transfer between agents of the same architecture, we train a set of five A2CV agents with different entropy costs and learning rates. To test transfer between agents with significantly different architectures, we train a set of five MERLIN-based agents (Wayne et al., 2018). These agents have a number of differences to the A2CV agents, most notably they contain a sophisticated memory structure based on a DNC (but with a fixed write location per timestep) (Graves et al., 2016). Both agents are trained on the same distribution and achieve human-level averages scores on the navigation task (with MERLIN scoring 10% higher than A2CV on average). Further details of agent training can be found in Appendix A.1.

To quantify the level of transfer between (sets of) agents, we follow the procedure for finding adversarial mazes outlined in Section 3.1 to produce a collection of 50 unique failure mazes for each agent (i.e. 10 collections of 50 mazes each). We then evaluate every agent 100 times on each maze in each collection, reporting their average performance on each collection. Complete quantitative transfer results can be found in Appendix A.4.

**Failure cases transfer somewhat across all agents** First, we find that across all agents, some level of transfer exists. In particular, as can be seen in Figure 6, the probability of one agent finding the goal on mazes generated to reduce the score another agent is significantly below 1. This suggests a common cause of failure that is some combination of the distribution of environment settings used during training and the set of methods that are currently used to train such agents. A possible way to address this could be enriching the training distribution so that it contains fewer biases and encourages more general solutions.

**Transfer within agent type is stronger than between agent type** Comparing the performance of each agent type on mazes from the same agent type to mazes from another agent type, we see that transfer within agent type is stronger. As shown in Figure 6b, performance increases as we go from 'MERLIN to MERLIN' to 'A2CV to MERLIN' (0.42 to 0.58) and also if we go from 'A2CV to A2CV' to 'MERLIN to A2CV' (0.63 to 0.70). This suggests that there are some common biases in agents that are due to their architecture type. Analyzing structural differences between mazes that lead to one agent type to fail but not another could give interesting insight into behavioural differences between agents beyond just average performance.

**A2CV agents are less susceptible to transfer** Despite similar probabilities of failure when evaluating on mazes optimized for the same agent, A2CV agents seem to suffer less on mazes optimized using other A2CV or MERLIN agents. This indicates that A2CV agents may have learned a more diverse set of strategies.

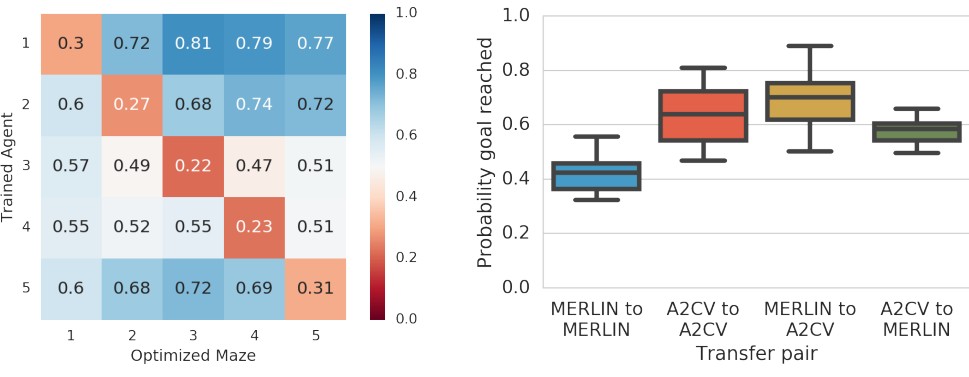

(a) Across hyperparameters, same architecture.                    (b) Across architectures.

Figure 6: **Mazes that lead to failures in one agent lead to failure in other agents as well.** This is the case for agents of the same architecture with different hyperparameters, and is also the case for transfer across agents of different architecture. We note, however, that transfer across agents with different architectures is weaker than among agents with the same architecture, and that the performance of agents with the same architecture but with different hyperparameters is slightly higher than for the agents used to originally find the mazes.

## 4    ADAPTING THE TRAINING DISTRIBUTION

From our experiments so far, we have discovered that there exist many mazes which lead to catastrophic failure. In this section, we investigate whether agent performance can be improved by adapting the training distribution, for example by incorporating adversarial mazes into training and modifying the original mazes used in training.

### 4.1    MOTIVATION

To better understand what may be causing catastrophic failures, with the aim of fixing them, we compare the set of adversarial mazes with the original set of mazes used in training. From this comparison, we find that there are two notable differences.

**The probability of a catastrophic failure correlates with the distance between the spawn locations and how hidden the goal is**  First, we find that a number of features are more common in adversarial mazes than non-adversarial mazes. In particular, adversarial mazes are more likely to have the goal hidden in an enclosed space (such as a small room), and on average the path length from the player's starting location to the goal is significantly longer ($31.1 \pm 8.4$ compared to $11.6 \pm 6.3$). Notably, while the training distribution also contains hidden goals which are far from the agent's starting location, they are much rarer.

**Adversarial mazes are typically far from the training distribution**  Second, we find that adversarial mazes tend to not only be out-of-distribution, but also far from the training distribution due to the `Modify` function used in our adversarial search procedure (for example, see Figure 2). This contrasts with the adversarial images literature where attacks are usually constrained to be small or imperceptible. It may therefore not be surprising that the agent is unable to generalize to all out-of-distribution mazes which could also explain the significant reduction in their performance.

Given these two observations, it is natural to ask whether the training distribution can be adapted to improve the agent's performance. In the following sections we investigate this question and discuss our findings, focusing on incorporating adversarial mazes into training and modifying the original mazes used in training.

## 4.2    APPROACH

We consider two distinct approaches for incorporating adversarial and out-of-distribution mazes into the training distribution.

**Adversarial training**  To add adversarial mazes into the training distribution, we first create a dataset of 6000 unique adversarial mazes from separate runs of our search procedure using the previously trained A2CV agents. Notably, this set also includes the 250 mazes used in our transfer experiments (Section 3.3). Next, we train a new set of A2CV agents using both this adversarial set of mazes and the standard distribution of mazes, sampling randomly every episode (i.e. 50% of training episodes are on an adversarial maze).

**Randomly perturbed training**  To ensure our adversarial search procedure produces in-distribution adversarial mazes, we alter the default maze generator used in training so that any adversarial maze can be generated. We accomplish this by randomly perturbing the original mazes, repeatedly using the same `Modify` function used by our adversarial search procedure, but selecting candidates randomly rather than by worst agent performance.

## 4.3    RESULTS

In this section, we report our findings on the robustness of agents trained using the approaches above.

**Catastrophic failures still exist**  Our main finding is that while agents learn to perform well on the richer distributions of mazes described above, this does not lead to robust agents. In particular, agents trained on a distribution of mazes enriched with 6000 adversarial mazes were able to find the goal on average 89.8% of the time on the adversarial mazes they were trained on. Similarly, agents trained on randomly perturbed mazes were able to find the goal close to 100% of the time on the distribution they were trained on. However, despite the agents being trained on these richer training distributions, the same search method is still able to find mazes leading to extreme failure as can be seen in Figure 7.

One possible explanation for this result is that the 6000 adversarial mazes used for training were insufficient to get good coverage of the space of mazes, and that further enlarging this set could yield qualitatively different results. Indeed, for agents trained using randomly perturbed mazes, the search procedure took 50 iterations to obtain the same level of failure as it did after 20 iterations when applied to agents trained on the standard training distribution. This suggests that perhaps enriching the training distribution with a very large set of adversarial mazes may lead to more general and robust agents. However, there are a number of challenges that need to be addressed before this approach can be tested which we will describe in the next section.

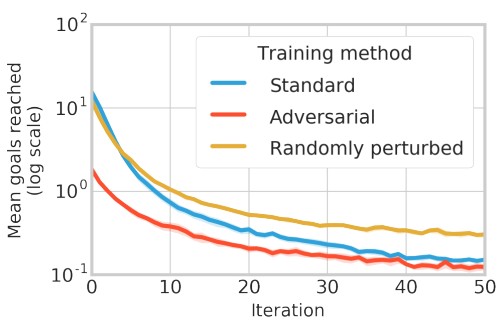 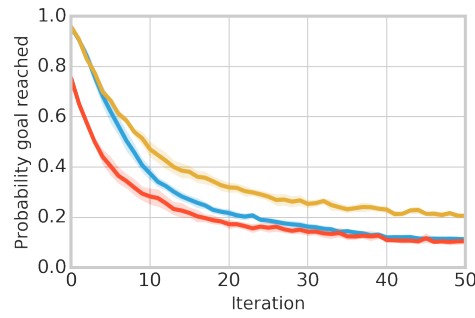

(a) Average number of goals reached.      (b) Probability of finding the goal in a single episode.

Figure 7: **Richer training distributions did not lead to robust agents.** Adversarial optimization for agents trained with adversarial mazes (red) and for agents trained with randomly perturbed mazes (yellow). Compared against the Standard training method from Figure 3 for 50 iterations (blue).

### 4.4 DISCUSSION

Our results suggest that if a richer training distribution is to yield more robust agents, we may need to use a very large set of environment settings leading to failure. This is similar to how adversarial training in supervised learning is performed where more adversarial examples are used than the original training examples. We describe below what we see as two significant challenges that need to be overcome before such an approach can be thoroughly evaluated in the RL setting.

**Expensive generation** The cost of generating a single adversarial setting is on the order of 1000's episodes using the method in this work. This implies that generating a set of adversarial settings which is similar in size to the set trained on would require orders of magnitude more computational than training itself. This could be addressed with faster methods for generating adversarial settings.

**Expensive training** Since agents receive very little reward in adversarial settings, the training signal is incredibly sparse. Therefore, it is possible that many more training iterations are necessary for agents to learn to perform well in each adversarial setting. A possible solution to this challenge is to design a curriculum over adversity, whereby easier variants of the adversarial settings are injected into the training distribution. For example, for the navigation tasks considered here, one could include training settings with challenging mazes where the goal is in any position on the shortest path between the starting location of the agent and the challenging goal.

We hope that these challenges can be overcome so that, in the context of RL, the utility of adversarial retraining can be established – an approach which has proved useful in supervised learning tasks. However, since significant challenges remain, we suspect that much effort and many pieces of work will be required before a conclusive answer is achieved.

## 5 RELATED WORK

**Navigation** Recently, there has been significant focus in the RL community on agent navigation in simulated 3D environments, including a community-wide challenge for agents in such environments called VizDoom (Kempka et al., 2016). Such 3D first-person navigation tasks are particularly interesting because they capture challenges such as partial observability, and require the agent to "effectively perceive, interpret, and learn the 3D world in order to make tactical and strategic decisions where to go and how to act." (Kempka et al., 2016). Recent advances have led to impressive human-level performance on navigation tasks in large procedurally generated environments (Beattie et al., 2016; Wayne et al., 2018).

**Adversarial examples in supervised learning** Our work can be seen as an RL navigation analogue of work on adversarial attacks on supervised learning systems for image classification (Szegedy et al., 2013). For adversarial attacks on image classifiers, one considers a set of inputs that

is larger than the original distribution, but where one would hope that systems perform just as well on $L_\infty$ balls around inputs from the distribution. In particular, the adversarial examples lie outside the training distribution. Analogously, we consider a set of mazes which is larger than the original set of mazes used during training, but where we would hope our system will work just as well.

Notably, while similar on a conceptual level, our setting has two key differences from this previous line of work: (1) The attack vector consists of changing latent semantic features of the environment (i.e. the wall structure of a maze), rather than changing individual pixels in an input image in an unconstrained manner. (2) The failure is realized over multiple steps of agent and environment interacting with each other, rather than simply being errant output from a single forward pass through a neural net.

More recently, in the context of supervised learning for image classification, there has been work to find constrained adversarial attacks which is closer to what we consider in this work (Athalye & Sutskever, 2017; Fawzi & Frossard, 2015; Eykholt et al., 2018; Sharif et al., 2016).

In the context of interpretable adversarial examples in image classification, similar approaches to our simplification approach have been explored where one searches for adversarial perturbations with group-sparse structure or other minimal structure (Xu et al., 2018; Brendel et al., 2018). Additionally, our findings regarding transfer are consistent with findings on adversarial examples for computer vision networks where it has been found that perturbations that are adversarial for one network often transfer across other networks (Szegedy et al., 2013; Tramèr et al., 2017)

**Input attacks on RL systems**    There have been a number of previous works which have extended adversarial attacks to RL settings, however they have achieved this by manipulating inputs directly, which effectively amounts to changing the environment renderer (Huang et al., 2017; Lin et al., 2017a;b). While these are interesting from a security perspective, it is less clear what they tell us about the generality of the strategy learned by the agent.

**Generalization in RL systems**    Recently, it has been shown that simple agents trained on restricted datasets fail to learn sufficiently general navigation strategies to improve goal retrieval times on held out mazes (Dhiman et al., 2018). In comparison, our method is both automatic and able to find more spectacular failures. Further, our findings highlight failures in *exploration* during navigation. This is in contrast to this previous work which studied failures to exploit knowledge from previous goal retrievals in the same episode.

In the context of testing generalization in RL, previous work has looked at statistical generalization in RL (Zhang et al., 2018). Here we consider agents that already generalize in the statistical sense and try to better characterize the ways in which they generalize beyond the average-case.

## 6 Conclusions and Future Work

In this work, we have shown that despite the strong average-case performance often reported of RL agents, worst-case analysis can uncover environment settings which agents have failed to generalize to. Notably, we have found that not only do catastrophic failures exist, but also that *simple* catastrophic failures exist which we would hope agents would have generalized to, and that failures also transfer between agents and architectures.

As agents are trained to perform increasingly complicated tasks in more sophisticated environments, for example AirSim (Shah et al., 2017) and CARLA (Dosovitskiy et al., 2017), it is of interest to understand their worst-case performance and modes of generalization. Further, in real world applications such as self-driving cars, industrial control, and robotics, searching over environment settings to investigate and address such behaviours is likely to be critical on the path to robust and generalizable agents.

To conclude, while this work has focused mostly on evaluation and understanding, it is only a first step towards the true goal of building more robust, general agents. Initial results we report indicate that enriching the training distribution with settings leading to failure may need to be done at a large scale if it is to work, which introduces significant challenges. While training robust agents is likely an endeavour requiring significant effort, we believe it is important if agents are to carry out critical tasks and on the path to finding more generally intelligent agents.

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

## APPENDIX A

### A.1  AGENT TRAINING

In this section we describe how the agents used in this work were trained.

#### A.1.1  A2CV

The A2CV agents in this paper are trained as in (Espeholt et al., 2018), but with a few modifications. We note that performance of the agents presented here is higher than that published in Espeholt et al. (2018). The differences in our training procedure were as follows:

- We train the agents for 10 billion steps as opposed to 333 million steps as in Espeholt et al. (2018).
- We use a simplified action set as in Hessel et al. (2018).
- We clip rewards to $[-1, 1]$.

The main cause of higher performance seems to be the approximately 30x increase in the number of training steps. Indeed, at 333 million steps, the agents trained here obtain a similar score to the agents in Espeholt et al. (2018). After training, the agents all achieved average rewards between 310 and 320 corresponding to finding the goal on average between 31 and 32 times per episode.

#### A.1.2  MERLIN

The agent model was a simplified variant of the model presented in Wayne et al. (2018), originally built to reduce training time in multi-task training scenarios. Specifically, the stochastic latent variable model was removed. This involved removing the prior network and directly producing a deterministic state representation using the same multi-layer perceptron as the posterior network in Wayne et al. (2018); however, instead of producing a Gaussian distribution and sampling, the state representation was a deterministic transformation $z_t = f(e_t, h_{t-1}, m_{t-1})$ as a function of the recurrent controller state and the read vectors retrieved at the previous time step from the external memory system. Additionally, the policy network was a purely feedforward multi-layer perceptron with one hidden layer of 200 units and a tanh nonlinearity computing the multinomial action distribution, also conditioned on the state representation $z_t$, recurrent state $h_t$, and memory reads $m_t$ at the current time step: $\pi(a_t|z_t, h_t, m_t)$. The policy loss was the same as for the A2CV agent. After training, the agents all achieved average rewards between 340 and 360 corresponding to finding the goal on average between 34 and 36 times per episode.

### A.2  ADVERSARIAL SEARCH PROCEDURE

#### A.2.1  COMPUTATIONAL REQUIREMENTS

As described in Figure 2, our search algorithm is ran using 10 candidate mazes per iteration, each evaluated 30 times, across 20 iterations. This is a total of 6000 episodes for the entire search procedure, and all episodes within one iteration can be evaluated in parallel (i.e. 20 batches of 300 episodes). In our experiments with 30 evaluations per maze, the entire search procedure took 30 minutes to complete, and only 9 minutes on average to find an adversarial maze where the probability of the agent finding the goal was below 50%. We also found reducing the number of evaluations per maze from 30 to 10 produced similar results and led to a 3x reduction in resources.

Our search procedure took around 30 minutes using 200 parallel workers each requiring approximately 2 CPUs . In contrast, agents were trained using 150 parallel workers each also requiring approximately 2 CPUs and taking 4 days.

#### A.2.2  ROBUSTNESS

In Figure 3 (Section 3.1), we report the average performance of 50 independent optimization runs (i.e. 50 different initializations of our search algorithm). In 44/50 (88%) of these runs, our search algorithm was able to find at least one adversarial maze where the agent's probability of finding the

goal was <50% (compared to 98% on the average maze). Furthermore, the 25th, 50th, and 75th percentiles were as follows:

- p(reaching the goal): 0.031, 0.136, 0.279
- number of goals reached: 0.042, 0.136, 0.368

### A.3 HUMAN EXPERIMENTS

To upper bound the intrinsic difficulty of the mazes found to be adversarial to agents, we conducted experiments where three humans played on the same mazes. Each human played a single episode on each of ten mazes. The humans played at the same resolution as agents, 96x72 pixels, to rule out visual acuity as a confounding factor. On all mazes, all humans successfully found the goal in the course of the episode. In fact, in most episodes, humans were able to find the goal in less than a third of the episode.

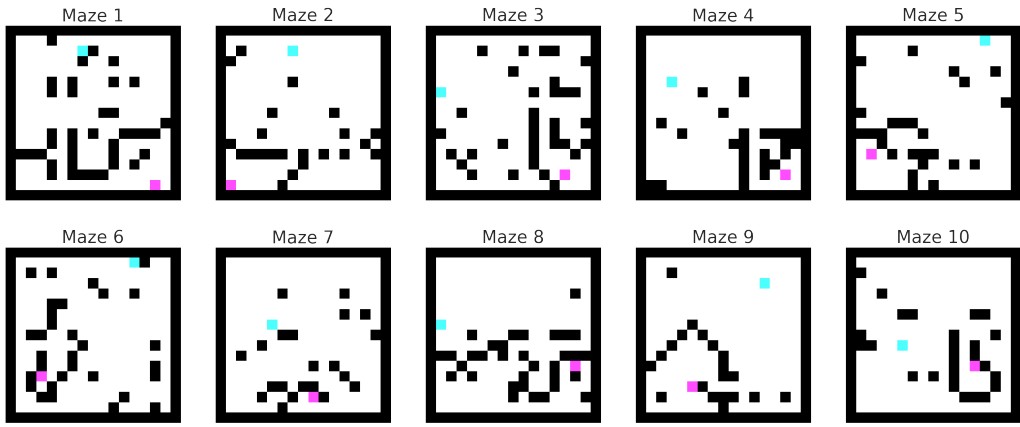

Figure 8: **Mazes used for human experiments.** For each maze, the agent that performed *best* found the goal less than 50% of the time. In contrast, humans always found the goal, usually within less than a third of the episode. Note that humans played at the same resolution as agents, 96x72 pixels.

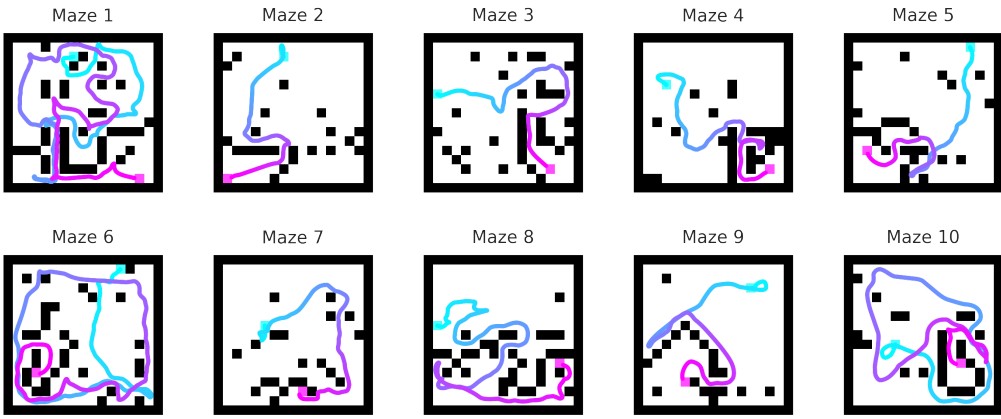

Figure 9: **Trajectories taken by Human 3 on mazes leading to agent failure.**

Table 1: Human seconds-to-first-goal on agent failure mazes

| Maze | 1 | 2 | 3 | 4 | 5 | 6 | 7 | 8 | 9 | 10 |
|---|---|---|---|---|---|---|---|---|---|---|
| **Human 1** | 13 | 14 | 27 | 15 | 18 | 47 | 41 | 37 | 24 | 16 |
| **Human 2** | 25 | 22 | 25 | 16 | 24 | 53 | 86 | 33 | 25 | 22 |
| **Human 3** | 64 | 14 | 16 | 18 | 17 | 52 | 17 | 33 | 22 | 42 |

## A.4 TRANSFER

In this section we provide detailed results for our transfer experiments. In particular, we detail transfer between all pairs among the 10 agents, five A2CV agents and five MERLIN agents trained with different entropy costs and learning rates.

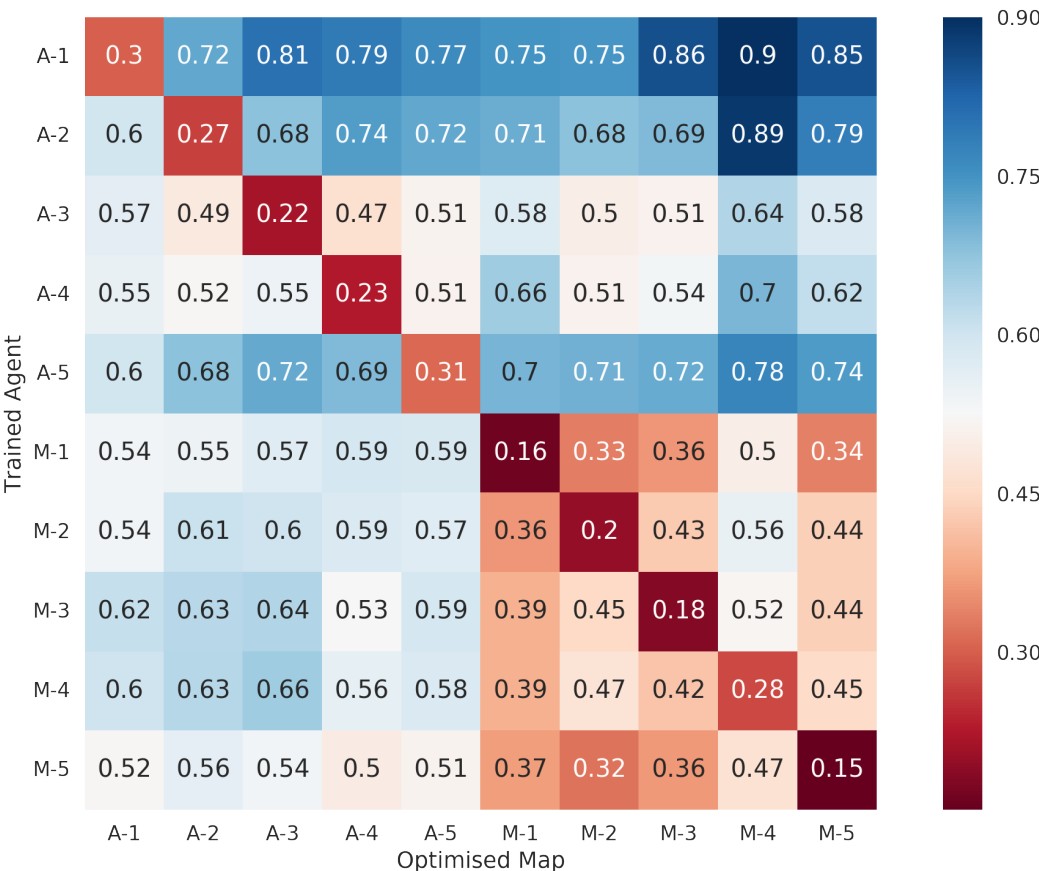

Figure 10: **Pairwise transfer scores.** Lower number indicates more transfer. 'A' corresponds to our A2CV agent, and 'M' corresponds to MERLIN.

