# OpenReview forum: "Uncovering Surprising Behaviors in Reinforcement Learning via Worst-case Analysis"
_ICLR.cc/2019/Conference_

### Official Review · AnonReviewer1 · 2018-10-27
**An interesting paper**

**Rating:** 6
**Confidence:** 2

**Review:**

This is an interesting paper, trying to find the adversarial cases in reinforcement learning agents. The paper discusses several different settings to investigate how generalizable the worst-case environment is across different models and conjectured that it comes from the bias in training the agents. Overall the paper is well-written and the experiments seem convincing. I have two questions regarding the presented result.

1. The search algorithm depicted in section 2 is only able to find a local optimum in the environment space. How robust is the result given different initializations?

2. It is briefly discussed in the paper that the failure in certain mazes might come from the structural bias in the training and the “complex” mazes are under-represented in the training dataset. It is hence natural to ask, if the procedure described in this paper can be incorporated to enhance the performance by some simple heuristics like re-weighting the training samples. I think some discussion on this would be beneficial for verifying the conjecture made here.

3. The authors compared the “hardness” of the mazes based on the number of walls in the maze. But it is arguably a good metric as the authors also mentioned visibility and other factors in measuring the complexity of the task. I would like to see more exploration in different factors that accounts for the complexity and maybe compare different agents to see if they are sensitive in the same set of factors.

To summarize, I like the idea of the paper and I think the result can be illuminating and worth some more follow-up work to understand the RL training in general.

---

> ### Author Response · Authors · 2018-11-14
> **Addressing questions on robustness, re-weighting training samples, and measures of complexity**
>
> Thank you for your positive comments. We hope our following responses address the three questions you raised.
>
> > 1. The search algorithm depicted in section 2 is only able to find a local optimum in the environment space. How robust is the result given different initializations?
>
> Great question. We found that our search algorithm is robust to different initialisations
>
> In Figure 3 (Section 3.1), we report the average performance of 50 independent optimisation runs (i.e. 50 different initialisations). Related to your question, in 44/50 (88%) of these runs our search algorithm found an adversarial maze where the agent’s probability of finding the goal was <50% (compared to 98% on the average maze). It is also possible to improve the robustness of our method by increasing the number of candidates considered per iteration (at the cost of increased time/computational requirements).
>
> The 25th, 50th, and 75th percentiles of our optimisation method were as follows:
>     - p(reaching the goal): 0.031, 0.136, 0.279
>     - number of goals reached: 0.042, 0.136, 0.368
>
> We will include a mention of these in the appendix.
>
> > 2. It is hence natural to ask, if the procedure described in this paper can be incorporated to enhance the performance by some simple heuristics like re-weighting the training samples.
>
> Please see the overall response to all reviewers for a full answer to this question.
>
> Summarising our full answer for your question - yes the procedure described in the paper can be used to re-weight the training samples, and we will include a section to the paper describing these experiments. However, we found that doing this is not sufficient to significantly enhance the performance of agents. Specifically, we found that re-weighting adversarial examples improved performance on those examples, but did not lead to agents improving overall.
>
> > 3. I would like to see more exploration in different factors that accounts for the complexity.
>
> This is an interesting point and relates to our motivation for adapting the training distribution. We investigated a number of different ways for measuring the complexity of mazes, comparing the distribution of various features between adversarial and non-adversarial mazes.
>
> Notably, we found differences in two measures of maze complexity: (1) the shortest path distance between the player’s start location and the goal location, and (2) the complexity of the shortest path to goal defined by the shortest path divided by the straight line distance to the goal. In particular, we found that adversarial mazes have a significantly longer shortest path to the goal on average, as well as a higher path complexity.
>
> These findings are in part what motivated us to adapt the training distribution, for example by re-weighting training samples which were adversarial. However, we observed minimal improvement doing this, and one of our hypotheses for this is that while the measures above are correlated with mazes being adversarial, they are not necessarily the cause (we discuss this more in our overall response to all reviewers).
>
> We will add these findings to the paper.

---

### Official Review · AnonReviewer3 · 2018-11-02

**Rating:** 7
**Confidence:** 4

**Review:**

Update:

I appreciate the clarifications and the extension of the paper in response to the reviews. I think it made the work stronger. The results in the newly added section are interesting and actually suggest that by putting more effort into training set design/augmentation, one could further robustify the agents, possibly up to the point where they do not break at all in unnatural ways. It is a pity the authors have not pushed the work to this point (and therefore the paper is not as great as it could be), but still I think it is a good paper that can be published.

-----

The paper analyzes the performance of modern reinforcement-learning-based navigation agents by searching for “adversarial” maze layouts in which the agents do not perform well. It turns out that such mazes  exist, and moreover, one can find even relatively simple maze configurations that are easily solved by humans, but very challenging for the algorithms.

Pros:
1) Interesting and relevant topic: it is important not only to push for best results on benchmarks, but also understand the limitations of existing approaches.
2) The paper is well written
3) The experiments are quite thorough and convincing. I especially appreciate that it is demonstrated that there exist simple mazes that can be easily solved by humans, but not by algorithms. The analysis of transferability of “adversarial” mazes between different agents is also a plus.

Cons:
1) I am not convinced worst-case performance is the most informative way to evaluate models. Almost no machine learning model is perfect, and therefore almost for any model it would be possible to find training or validation samples on which it does not perform well. Why is it so surprising that this is also the case for navigation models? Why would one assume they should be perfect? Especially given that the generated “adversarial” mazed lie outside of the training data distribution, seemingly quite far outside. Are machine learning models ever expected to perfectly generalize outside of the training data distribution? Very roughly speaking, the key finding of the paper can be summarized as “several recent navigation agents have problems finding and entering small rooms of the type they never saw during training” - is this all that significant?

To me, the most interesting part of the paper is that the models generalize as well as they do. I would therefore like to see if it is possible to modify the training distribution - by adding “adversarial” mazes, potentially in an iterative fashion, or just by hand-designing a wider distribution of mazes - so that generalization becomes nearly perfect and the proposed search method is not anymore able to find “adversarial” mazes that are difficult for the algorithm, but easy for humans.

2) On a related note, to me the largest difference between the mazes generated in this paper and the classical adversarial images is that the modification of the maze is not constrained to be small or imperceptible. In fact, it is quite huge - the generated mazes are far from the training distribution. This is a completely different regime. This is like training a model to classify cartoon images and then asking it to generalize to real images (or perhaps other way round). Noone would expect existing image classification models to do this. This major difference with classical adversarial examples should be clearly acknowledged.

3) It would be interesting to know the computational requirements of the search method. I guess it can be estimated from the information in the paper, but would be great to mention it explicitly. (I am sorry if it is already mentioned and I missed it)

To conclude, I think the paper is interesting and well executed, but the presented results are very much to be expected. To me the most interesting aspect of the work is that the navigation agents generalize surprisingly well. Therefore, I believe the work would be much more useful if it focused more on how to make the agents generalize even better, especially since there is a very straightforward way to try this - by extending the training set. I am currently in the borderline mode, but would be very happy to change my evaluation if the focus of the paper is somewhat changed and additional experiments on improving generalization (or some other experiments, but making the results a bit more useful/surprising) are added.

---

> ### Author Response · Authors · 2018-11-14
> **Addressing questions on perfection, modifying the training distribution, and the computational requirements of the search method**
>
> Thank you for your detailed comments. We hope our responses below address your comments.
>
> > 1) Almost no machine learning model is perfect... Why is it so surprising that this is also the case for navigation models? Why would one assume they should be perfect?
>
> We agree that it is not necessarily surprising that these failure cases exist, and this is not the key result we wanted to highlight as surprising. While surprise is clearly subjective, we, and others we spoke to, found the following results surprising: (1) lack of generalisation to simpler situations, (2) how extreme the failures are (not just that the algorithm is imperfect, but 100x’s reduction in performance), (3) transfer of failures between different agents/architectures (so these failures aren’t just overly-specialised to specific training runs).
>
> > 1) I would therefore like to see if it is possible to modify the training distribution - by adding “adversarial” mazes.
>
> Please see the overall response to all reviewers for a full answer to this question.
>
> Summarising our full answer for your question - yes it is possible to modify the training distribution. However, in the case of adding “adversarial” mazes to the training distribution, we found that while agents learned to perform well on the specific mazes added, they did not perform better on adversarial mazes overall. It is possible that adding many more adversarial mazes into the distribution will lead to more robustness (as is the case for some image classification datasets such as MNIST and CIFAR10), however this would require significant new techniques for finding adversarial mazes more efficiently, and this is another line of inquiry we are currently pursuing.
>
> > 2) The modification of the maze is not constrained to be small or imperceptible. In fact, it is quite huge - the generated mazes are far from the training distribution … . This major difference with classical adversarial examples should be clearly acknowledged.
>
> Please see the overall response to all reviewers for a full answer to this question.
>
> To address this concern, we have altered the maze generator so that any adversarial maze can be generated and seen during training, meaning our adversarial search procedure produces in-distribution adversarial mazes. We accomplished this by randomly altering the original mazes using the same Modify function used by our adversarial search procedure. The result of this is that the generated mazes are no longer far from the training distribution.
>
> We will add an acknowledgement to the difference between our current analysis and the classical adversarial examples, and highlight how our new section on adapting the training distribution helps address this point.
>
> > 3) It would be interesting to know the computational requirements of the search method.
>
> Our search method is performed using 10 candidate mazes per iteration, each evaluated 30 times, across 20 iterations. This is a total of 6000 episodes for the entire search procedure, and all episodes within one iteration can be evaluated in parallel (i.e. 20 batches of 300 episodes).  Depending on the desired confidence level and resources available, the number of evaluations can be increased or decreased. Indeed, we found that evaluating each maze only 10 times rather than 30 produced similar results and led to a 3x speed up. In our experiments with 30 evaluations per maze, the entire search procedure took 30 minutes to complete, and only 9 minutes on average to find an adversarial maze where the probability of the agent finding the goal was below 50%. As suggested, we will add an explicit mention of this in the paper.

---

### Official Review · AnonReviewer2 · 2018-11-02
**A simple idea with interesting results, but lacking in broader impact**

**Rating:** 5
**Confidence:** 3

**Review:**

The authors present a simple technique for finding "worst-case" maze environments that result in bad performance. The adversarial optimization procedure is a greedy procedure, which alternately perturbs maze environments and selects the maze on which the trained agent performs worst for the next iteration. The authors highlight three properties of these mazes, which show how this adversarial optimization procedure can negatively impact performance.

High-level comments:
- I am unconvinced that many of the observed behaviors are "surprising". The procedure for adversarially optimizing the maps is creating out-of-distribution map samples (this is confirmed by the authors). The function for creating maps built-in to DeepMind Lab (the tool used to generate the random maps used in this paper) has a set of rules it uses to ensure that the map follows certain criteria. Visual inspection of the 'Iteration 20' maps in Figure 2 finds that the resulting adversarial map looks fundamentally different from the 'Initial Candidate' maps. As a result, many of the features present in the adversarial maps may not exist in the initial distribution, and the lack of generalizability of Deep RL has become a relatively common talking point within the community. That being said, I agree with the authors' claims about how this sort of analysis is important (I discuss this more in my second point).
- In my mind, the 'discovery' of the performance on these optimized out-of-distribution samples is, in my mind, not particularly impactful on its own. The Deep RL community is already rather aware of the lack of generalization ability for agents, but are in need of tools to make the agents more robust to these sorts of examples. For comparison, there is a community which researches techniques to robustify supervised learning systems to adversarial examples (this is also mentioned by the authors in the paper). I feel that this paper is only partially complete without an investigation of how these out-of-distribution samples can be used to improve the performance of the agents. The addition of such an investigation has the potential to greatly strengthen the paper. This lack of "significance" is the biggest factor in my decision.
- The first two results sections outlining the properties of the adversarially optimized mazes were all well-written and interesting. While generally interesting, that the less-complex A2CV agent shows better generalization performance than the more-complex MERLIN agent is also not overly surprising. Yet, it remains a good study of a phenomenon I would not have thought to investigate.

Minor comments:
- The paper is very clear in general. It was a pleasure to read, so thank you! The introduction is particularly engaging, and I found myself nodding along while
- Figures are generally excellent; your figure titles are also extremely informative, so good work here.
- Fig 4. It might be clearer to say "adversarially optimized" instead of simplly "optimized" in the (b) caption to be clearer that it the map that is being changed here, rather than the agent. Also, "Human Trajectories" -> "Human Trajectory", since there is only one.
- I am not a fan of saying "3D navigation tasks" for 2.5D environments (but this has become standard, so feel free to leave this unchanged).

This paper is a well-written investigation of adversarially chosen out-of-distribution samples. However, the the high-quality of this narrow investigation still only paints a partial picture of the problem the authors set out to address. At the moment, I am hesitant to recommend this paper for acceptance, due to its relatively low "significance"; a more thorough investigation of how these out-of-distribution samples can be used.

---

> ### Author Response · Authors · 2018-11-14
> **Addressing comments on "surprise" and out-of-distribution samples**
>
> Thank you for your constructive comments. We’re glad you enjoyed reading the paper and hope our responses below address your concerns.
>
> Please see the overall response to all reviewers for a detailed answer on adapting the training distribution which we will include in the paper. Here we will provide answers specific to the points you raised.
>
> > I am unconvinced that many of the observed behaviors are "surprising".
>
> While surprise is clearly subjective, we, and others we spoke to, found the following results surprising: (1) lack of generalisation to simpler situations, (2) how extreme the failures are (not just that the algorithm is imperfect, but 100x’s reduction in performance), (3) transfer of failures between different agents/architectures (so these failures aren’t just overly-specialised to specific training runs). Importantly, we found that these (arguably surprising) behaviours were also present after we addressed the out-of-distribution concern which we explain in the next point.
>
> > The procedure for adversarially optimizing the maps is creating out-of-distribution map samples.
>
> To address this, we have altered the maze generator so that any adversarial maze can be generated and seen during training, meaning our adversarial search procedure produces in-distribution adversarial mazes. We accomplished this by randomly altering the original mazes using the same Modify function used by our adversarial search procedure.
>
> The result of this change is that our adversarial search procedure takes several more iterations to find a maze as-adversarial as before, however it is still possible to find such mazes and there is no significant improvement in agent performance.
>
> > I feel that this paper is only partially complete without an investigation of how these out-of-distribution samples can be used to improve the performance of the agents.
>
> We agree, and the other reviewers also mentioned this, therefore we will be adding a section on this topic to the paper. We found that using these out-of-distribution samples to improve the performance of the agents is not straightforward, and the approaches we tried (which were also suggested by various reviewers) were not sufficient for significantly improving performance.
>
> In the case of adding adversarial samples to training, we found that agents learned to solve the specific adversarial samples they were trained on, but did not perform better on adversarial samples overall.

---

### Author Response · Authors · 2018-11-14
**Overall response to all reviewers on out-of-distribution samples and adapting the training distribution**

As adapting the training distribution was a common point raised by all reviewers, we provide this overall response for all reviewers.

We absolutely agree that adapting the training distribution is an important direction, and during the work for this paper we investigated several methods for incorporating out-of-distribution mazes into the training distribution.

Specifically, we tried two approaches related to what the reviewers suggested:
    - Altered Training: we altered the maze generator so that any adversarial maze could be generated and seen during training, meaning our adversarial search procedure produces in-distribution adversarial mazes. We accomplished this by randomly altering the original mazes, repeatedly using the same Modify function used by our adversarial search procedure, but without selecting for worst agent performance.
    - Adversarial Training: we incorporated a large set of adversarial mazes into the training distribution, including the 500 mazes we used in our transfer analysis experiment. This was achieved by having two sets of mazes (the default distribution and the adversarial distribution) which were then randomly sampled every episode (i.e. 50% of training episodes were on an adversarial maze).

To summarise, we found that -- perhaps surprisingly -- neither of these approaches significantly improved the agent’s robustness to our adversarial search procedure (i.e. numerous adversarial mazes still exist and can be found), and therefore adapting the training distribution in such a way that improves performance in the dimension we are interested in is more challenging than initially thought.

Given this result, we originally did not include these experiments as we felt they were too preliminary and the topic was worth a thorough investigation in subsequent work. However, given the surprise of the result, the further work we have put into these results since submission, and that all reviewers suggested trying it, we will include an additional section in the paper on ‘Adapting the Training Distribution’ where we will include our experimental results.

We list the two key results here while we update the paper:
    - Altered Training: we found that training on altered mazes made it marginally harder for our search method to find adversarial mazes (i.e. it took several more iterations), however overall this did not fix the problem.
    - Adversarial Training: we found that incorporating adversarial examples into the training distribution led to agents performing well on those specific examples, however this did not lead to agents performing better against our adversarial search method.

For Adversarial Training, it is possible that adding many more adversarial mazes into the distribution will lead to more robustness (as is the case for some image classification datasets such as MNIST and CIFAR10). However, this would require significant new techniques for finding adversarial mazes more efficiently, and is another line of inquiry we are currently pursuing.

We also found that there were quantifiable differences between the adversarial and non-adversarial distribution of mazes (e.g. goals in small rooms, long path from the player to the goal). Indeed, as all reviewers mentioned, this motivates adding out-of-distribution / rare mazes into the training distribution. However, importantly, we note that while we identified features which correlated with adversarial mazes, there is not necessarily causation. To investigate causation, we tried handcrafting mazes with the identified features of the adversarial distribution, however we were not able to consistently create mazes which were adversarial. This point of correlation versus causation is one possible reason why adapting the training distribution is more challenging than initially thought.

We believe the addition of this new section will provide a more complete investigation while also opening up a number of interesting research directions for future work. We believe it is likely that, similar to adversarial examples in image classification domains, figuring out how to train agents to be more robust and general is likely to take significant time and effort, and therefore span many papers and works.

---

### Author Response · Authors · 2018-11-22
**Paper Updated**

Dear Reviewers,

Thank you for the constructive feedback and suggestions. We have now updated the paper based on your comments.

Most notably, we have updated the paper to include our experiments on adapting the training distribution using adversarial and out-of-distribution examples (see new Section 4). This addresses a majority of the questions and concerns raised by all reviewers, and follows what we mentioned in our previous comments. For example, our adversarial search procedure now finds in-distribution examples and we also present results on incorporating adversarial examples into the training distribution.

Additionally, we have added details on the computational requirements and robustness of our adversarial search procedure to the appendix (Appendix A.2). We have also made minor changes throughout to improve the readability of the paper.

We believe these updates address the comments made by the reviewers. If the reviewers are satisfied by our responses, we hope they will consider raising their scores or letting us know in what ways they think the paper should be improved before the discussion period ends.

Thanks,
Authors

---

### Meta-Review · Area_Chair1 · 2018-12-15
**Very interesting approach, unsure about the usefulness in the current state of experiments**

**Confidence:** 3
**Recommendation:** Reject

**Metareview:**

The paper presents adversarial "attacks" to maze generation for RL agents trained to perform 2D navigation tasks in 3D environments (DM Lab).

The paper is well written, and the rebuttal(s) and additional experiments (section 4) make the paper better. The approach itself is very interesting. However, there are a few limitations, and thus I am very borderline on this submission:
 - the analysis of why and how the navigation trained models fail, is rather succinct. Analyzing what happens on the model side (not just the features of the adversarial mazes vs. training mazes) would make the paper stronger.
 - (more importantly) Section 4: "adapting the training distribution" by incorporating adversarial mazes into training feels incomplete. That is a pithy as giving an adversarial attack for RL trained navigation agents would be much more complete of a contribution if at least the most obvious way to defend the attack was studied in depth. The authors themselves are honest about it and write "Therefore, it is possible that many more training iterations are necessary for agents to learn to perform well in each adversarial setting." (under 4.4 / Expensive Training).

I would invite the authors to submit this version to the workshop track, and/or to finish the work started in Section 4 and make it a strong paper.